# Experimental Study of Polypropylene with Additives of Bi_2_O_3_ Nanoparticles as Radiation-Shielding Materials

**DOI:** 10.3390/polym14112253

**Published:** 2022-05-31

**Authors:** Ahmed M. El-Khatib, Thanaa I. Shalaby, Ali Antar, Mohamed Elsafi

**Affiliations:** 1Physics Department, Faculty of Science, Alexandria University, Alexandria 21511, Egypt; elkhatib60@yahoo.com; 2Department of Medical Biophysics, Medical Research Institute, Alexandria University, Alexandria 21561, Egypt; th_shalaby@yahoo.com (T.I.S.); newoten84@yahoo.com (A.A.)

**Keywords:** polypropylene, Bi_2_O_3_ nanoparticles, SEM, mechanical, radiation shielding

## Abstract

This work aimed to intensively study polypropylene samples (PP) embedded with micro- and nanoparticles of Bi_2_O_3_ for their application in radiation shielding. Samples were prepared by adding 10%, 20%, 30%, 40%, and 50% of Bi_2_O_3_ microparticles (mBi_2_O_3_) by weight, and adding 10% and 50% of Bi_2_O_3_ nanoparticles (nBi_2_O_3_), in addition to the control sample (pure polypropylene). The morphology of the prepared samples was tested, and also, the shielding efficiency of gamma rays was tested for different sources with different energies. The experimental LAC were determined using a NaI scintillation detector, the experimental results were compared with NIST-XCOM results, and a good agreement was noticed. The LAC values have been used to calculate some specific parameters, such as half value layer (HVL), mean free path (MFP), tenth value layer (TVL), and radiation protection efficiency (RPE), which are useful for discussing the shielding capabilities of gamma rays. The results of the shielding parameters show that the PP embedded with nBi_2_O_3_ gives better attenuation than its counterpart, PP embedded with mBi_2_O_3_, at all studied energies.

## 1. Introduction 

Nowadays, man-made sources of radiation range in diversity, from nuclear power plants to the medical uses of radiation in diagnosing diseases or treating patients. It was found that the most common man-made sources of ionizing radiation are radioisotopes, X-ray machines, and other medical devices used in hospitals, oncology centers, and the medical industry [1,2,3]. As standards have evolved, the general approach has been to rely on risk estimates that have little chance of underestimating the consequences of radiation exposure, and to estimate the risks in different occupational environments associated with radiation exposure; however, it is important to understand the biological effects of radiation exposure [4,5,6].

Shielding is one of the most important factors, as materials are used to absorb and attenuate radiation, and are used, to an appropriate extent, to reduce the amount of radiation [7,8,9,10]. Lead, bismuth, and concrete are among the most important materials used in minimizing the penetration of ionizing radiation, and as a result of the great developments in the field of nanotechnology, many researchers are working to synthesize many inexpensive materials, such as glass and polymer, and even their waste, to enhance their properties by adding and mixing nanoparticles, such as lead and bismuth, to work as a highly efficient shield against radiation from X-ray medical devices and radioactive sources [11,12,13,14].

Polypropylene is an economical material that offers a combination of outstanding physical, chemical, mechanical, thermal, and electrical properties not found in many other thermoplastic materials [15]. Polypropylene is characterized by light weight, high tensile strength, impact resistance, high pressure resistance, excellent insulating properties, resistance to most acids and alkalis, resistance to stress cracking, maintaining toughness and elasticity, low moisture absorption, non-toxicity, easy fabrication, and high heat resistance [16,17].

Researchers have developed and improved the properties of many materials for use in radiation shielding. Polymer composites are reinforced by metal oxides such as bismuth dioxide, which is the most used filler in polymeric matrices to shield gamma rays due to its high density and high atomic number compared to other metal oxides [18,19,20,21,22]. The role of polymer is to acquire plasticity, have an easy formability, and to provide load–stress transfer.

This work gives attention to polymer composites of recycled waste polypropylene as a radiation shield. The prepared composites were filled with powdered bulk bismuth dioxide and bismuth dioxide nanoparticles with different percentage filler weight fractions. Moreover, this study aimed to evaluate the ability of PP-Bi_2_O_3_ versus the PP-Bi_2_O_3_ NP_S_ in attenuating gamma rays.

## 2. Materials

### 2.1. Polypropylene (PP)

Polypropylene is an economical material that offers a combination of outstanding physical, chemical, mechanical, thermal, and electrical properties not found in any other thermoplastic material. Compared with low- or high-density polyethylene, it has a lower impact strength, but superior working temperature and tensile strength. Its features are light weight, high tensile strength, impact resistance, high pressure resistance, excellent insulating properties, and non-toxicity. Its density ranges from 0.901 to 0.905 g/cm^−3^, its tensile strength is 4800 psi, its tensile modulus is 195,000 psi, its tensile elongation at yield is about 12%, the compressive strength is 7000 psi, and the Rockwell hardness test is 92 [23]. It was collected from Sidi Kerir Petrochemical Company in Egypt, with a melting flow point index of 0.38 g/min and a density of 0.902 g/cm^3^.

### 2.2. Bismuth Oxide (Bi_2_O_3_)

In this work, micro- and nano-sized bismuth oxide particles were used as fillers. Microparticles were purchased locally from Abico Pharmaceuticals, with a purity of 98.9% and an average size of about 100 μm, whereas nanoparticles were purchased from Nano Tech Company, as they were chemically prepared. 

### 2.3. Polymer Mix Design 

The samples in this study were prepared using a pressure-molding method for all polymer samples, as shown in Table 1. First, a 0.0001 g sensitive electrostatic balance was used to weigh waste polypropylene and bismuth oxide, and then, PP was placed into a cylindrical mill at 165 °C (which is above the melting point of polypropylene) for 20 min at a rotational speed of 40 rpm. After the polypropylene was completely melted, the Bi_2_O_3_ powder, whether micro or nano, was added gradually with continuous rolling for 15 min to reach a uniform distribution of the powder in PP. The whole mixed sample was placed in an iron frame with dimensions of 12.5 × 12.5 × 3 cm. Then, the samples were compressed by a hydraulic heat press at a pressure of 10 MPa and a temperature of 850 °C for 15 min, and the pressure was gradually raised to 20 MPa for another 15 min. The sample was kept under pressure for 30 min to cool down gradually to a temperature of 400 °C, after which, the pressure sample was taken and cut into circular discs for measurement [24].

## 3. Methodology 

### 3.1. Morphological Test

Scanning electron microscope or SEM analysis (JSM-6010LV, JEOL Ltd., Tokyo, Japan) was used to monitor the distribution, size, and difference of micro and Bi_2_O_3_ NPs in the prepared composites. Images were acquired from SEM at a magnification order of 5000× at 20 kV [25].

### 3.2. Radiation Shielding Test 

Sodium iodide scintillation detector (NaI) and different radioactive point sources were used to test the attenuation parameters of the prepared samples [26,27]. Each prepared sample was tested for three different thicknesses, 0.5, 1.5, and 2 cm, with a fixed diameter of 8 cm. At first, the detector was calibrated (energy and efficiency calibration). The measurements were carried out at a fixed geometry where the distance between the source and the tested composite sample with thickness (*t*, cm) and density (ρ, g/cm^3^) was 24 cm, whereas the distance between the tested sample and the detector was 4 cm, as shown in Figure 1. The collected spectra were analyzed using the Genie software program. The net area per unit time for each energy peak in the spectrum (*N*_0_) and (*N*) for a particular radioactive source was determined in the absence and in the presence of the tested composite sample. The characteristics of the radioactive sources used in the measurements are listed in Table 2 [28,29].

To know the shielding ability of the material, the linear attenuation coefficient (LAC) was experimentally determined from the following equation [30]:(1)$$LAC=\frac{1}{t}ln\frac{N_{0}}{N}$$

To confirm the accuracy of the experimental measurements, the experimental results of *LAC* for PP-m Bi_2_O_3_ samples were compared with the results obtained from NIST XCOM. The linear attenuation coefficient (*LAC*) is the probability of photon interaction with polymer sample per unit path-length.

The half and tenth value layers (*HVL* and *TVL*) are the material thicknesses enough to reduce the gamma ray intensity by 50% and 10% of its initial intensity, respectively, whereas the mean free path (*MFP*) is defined as the average distance between two successive collisions. These parameters were calculated by the following equation [31,32]:(2)$$HVL=\frac{LN\left(2\right)}{LAC},\;TVL=\frac{LN\left(10\right)}{LAC},\;MFP=\frac{1}{LAC}$$

The radiation protection efficiency (*RPE*) is an important parameter for estimating the efficacy of shielding materials [33,34].
(3)$$RPE,\%=\left[1-\frac{N}{N_{0}}\right]\times 100$$

## 4. Results and Discussion

### 4.1. TEM and SEM Results

Transmission electron microscopy (TEM) (JEM-2100F, JEOL, Japan) at 200 kV was performed, as seen in Figure 2. By examining these characteristics, it was confirmed that the average size of Bi_2_O_3_ NPs was 20 ± 5 nm. The prepared samples of PP-m Bi_2_O_3_ and PP-n Bi_2_O_3_ were examined using scanning electron microscopy (SEM) to investigate the particle distribution inside the polypropylene, in addition to their sizes, as shown in Figure 3. It turns out that the distribution of nanoparticles is more diffuse than fine particles: the smaller the size of the Bi_2_O_3_ particles, the greater their spread, and they are more homogeneous inside the polymer. A material with this structure has less porosity, and works to attenuate the radiation with higher efficiency.

### 4.2. Attenuation Results

The LAC for free PP and PP-mBi_2_O_3_ composite samples were experimentally determined and compared with the results obtained from the NIST-XCOM software. The relation between both results was graphed in Figure 4, and R^2^ were estimated from each graph to show the agreement percentage for each one. The experimental results were plotted in the *y*-axis, whereas the theoretical results were plotted in the *x*-axis for all synthesized PP samples embedded with micro Bi_2_O_3_. The values of R^2^ were 0.9998, 0.9998, 0.9998, 0.9999, 0.9998, and 0.9997 for PP, PP-10m Bi_2_O_3_, PP-20m Bi_2_O_3_, PP-30m Bi_2_O_3_, PP-40m Bi_2_O_3_, and PP-50m Bi_2_O_3_, respectively. The LAC was calculated at different energies, and the results showed the impact of the added bismuth oxide on the remarkable increase in the attenuation coefficient, as depicted in Figure 5. Figure 5 shows that as the photon energy increases, the attenuation coefficient decreases for all discussed samples, and on the other hand, PP-50m Bi_2_O_3_ has the highest attenuation at all studied energies, whereas PP has lowest attenuation. At 0.060 MeV, the LAC was 0.1806, 0.6617, 1.2481, 1.9786, 2.9129, and 4.1513 cm^−1^ for PP, PP-10m Bi_2_O_3_, PP-20m Bi_2_O_3_, PP-30m Bi_2_O_3_, PP-40m Bi_2_O_3_, and PP-50m Bi_2_O_3_, respectively, whereas these samples have values of 0.0614, 0.0670, 0.0737, 0.0822, 0.0930, and 0.1073 cm^−1^, respectively, at 1.173 MeV.

These values indicated a good agreement between the experimental and theoretical results, as shown in Figure 4. This indicates the validity of the experimental setup, and, from this point, it was worthwhile to find the values of LAC for PP composites with Bi_2_O_3_ NPs (PP-10n Bi_2_O_3_ and PP-50n Bi_2_O_3_) experimentally. The LAC was measured for two samples containing Bi_2_O_3_ NPs, PP-10n Bi_2_O_3_, and PP-50n Bi_2_O_3_, and compared with the corresponding PP-m Bi_2_O_3_ composites. Figure 6a displays the comparison between PP-10m Bi_2_O_3_ and PP-10n Bi_2_O_3_ of the LAC results. The results showed a clear superiority of nanoparticles as a filler in polypropylene in all the studied energies, for example, at 0.081 MeV, the LAC was 0.3861 cm^−1^ for PP-10m Bi_2_O_3_, while being 0.4507 cm^−1^ for PP-10n Bi_2_O_3_, and the LAC was 0.0623 cm^−1^ for PP-10m Bi_2_O_3_, while being 0.0675 cm^−1^ for PP-10n Bi_2_O_3_ at 1.333 MeV. Similarly, The LAC results for PP-50m Bi_2_O_3_ and PP-50n Bi_2_O_3_ were plotted in Figure 6b. Here, the superiority was very noticeable over the previous 10% of bismuth oxide, where at 0.081 MeV, the LAC was 1.9821 cm^−1^ for PP-10m Bi_2_O_3_, while being 2.6186 cm^−1^ for PP-10n Bi_2_O_3_, and the LAC was 0.0990 cm^−1^ for PP-10m Bi_2_O_3_, while being 0.1154 cm^−1^ for PP-10n Bi_2_O_3_ at 1.333 MeV. The ratio between the micro and nano filler in polypropylene was calculated and graphed in Figure 6c for PP-10 Bi_2_O_3_ and PP-50 Bi_2_O_3_. The ratios in the PP-10 Bi_2_O_3_ sample were plotted at different energies and were 1.178, 1.1672, 1.138, 1.112, 1.090, and 1.083 at 0.060, 0.081, 0.356, 0.662, 1.173, and 1.333 MeV, respectively, which means that the nano/micro filler ratio is greater than 1, and the ratio decreases when increasing the energy, approaching 1 at high energy. Similarly, the ratio between the micro and nano filler for PP-50 Bi_2_O_3_ was 1.343, 1.322, 1.282, 1.237, 1.181, and 1.165 at 0.060, 0.081, 0.356, 0.662, 1.173, and 1.333 MeV, respectively. The ratio in PP-50 Bi_2_O_3_ was greater than the ratio in PP-10 Bi_2_O_3_, which is because the distribution of nanoparticles inside the polymer was more homogenous than the microparticles of Bi_2_O_3_. Similarly, the relative deviations of the micro and nano filler in polypropylene were calculated in Figure 6d for PP-10 Bi_2_O_3_ and PP-50 Bi_2_O_3_. The greatest deviation was 34.3% for PP-50 Bi_2_O_3_ at 0.060 MeV, whereas the greatest deviation for PP-10 Bi_2_O_3_ was 17.9% at the same energy. In contrast, The lowest deviation was 16.4% for PP-50 Bi_2_O_3_ at 1.333 MeV, whereas the lowest deviation for PP-10 Bi_2_O_3_ was 3.8% at 1.333 MeV.

The important attenuation parameters based on LAC calculation, such as the HVL, MFP, and TVL, were calculated for PP-10m Bi_2_O_3_, PP-10n Bi_2_O_3_, PP-50m Bi_2_O_3_, and PP-50n Bi_2_O_3_ at different energies, and tabulated in Table 3. The results indicated that nanoparticles filler gives an advantage over its microparticles counterpart in all attenuation coefficients, and the reason for this is that nanoparticles give a higher surface area and better distribution inside polypropylene.

The efficiency of the prepared materials for attenuation were calculated by radiation protection efficiency law, as shown in Figure 7. The values of RPE decrease when increasing the energy for all prepared samples, and the sample with the lowest RPE was PP-10m Bi_2_O_3_, whereas the samples with the highest RPE were PP-50n Bi_2_O_3_ at all discussed energies. The nano samples have RPE values that are superior to that of the micro samples, except at the low-studied energies (0.060 and 0.081 MeV), and when 50% of both micro and nano Bi_2_O_3_ are incorporated into polypropylene, the RPE values reach almost 100%, as shown in Figure 7. After that, the RPE values gradually decrease with the increase of energy for all the studied samples. For example, the sample PP-50n Bi_2_O_3_ has values of 100.00%, 99.69%, 71.11%, 45.41%, 31.52%, and 29.25% at energies of 0.060, 0.081, 0.356, 0.662, 1.173, and 1.333 MeV, respectively.

## 5. Conclusions

Polypropylene (PP) samples embedded with Bi_2_O_3_ micro- and nanoparticles were extensively studied for their application in radiation attenuation. The morphological test was carried out using SEM for the prepared samples, and it was found that the addition of nanoparticles improves the morphological properties and reduces the voids in the polymer compared to the microparticles. On the other hand, the protection efficiency of gamma rays was tested for different sources with different energies. The experimental LAC was determined using the NaI detector, and the experimental results were compared with those of NIST-XCOM, and a good agreement was observed. The results of the shielding parameters show that PP embedded with nano Bi_2_O_3_ gives better attenuation than that of PP embedded with micro Bi_2_O_3_ at all studied energies. From the foregoing, we conclude that these materials can be used in many applications, including the preservation of liquid radioactive sources in plastic materials made of this polymer. In addition, it can be used as an additional protective shield on walls, doors, and windows.

## Figures and Tables

**Figure 1 polymers-14-02253-f001:**
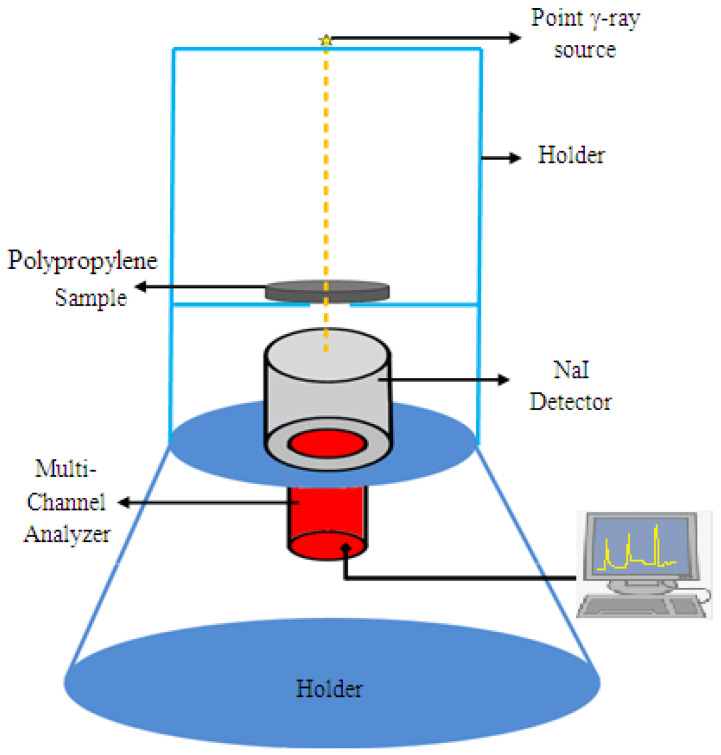
The illustration setup of the experimental work.

**Figure 2 polymers-14-02253-f002:**
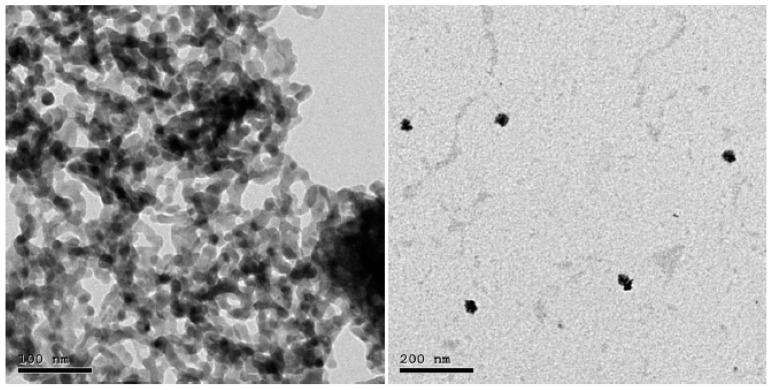
TEM images of Bi_2_O_3_ nanoparticles.

**Figure 3 polymers-14-02253-f003:**
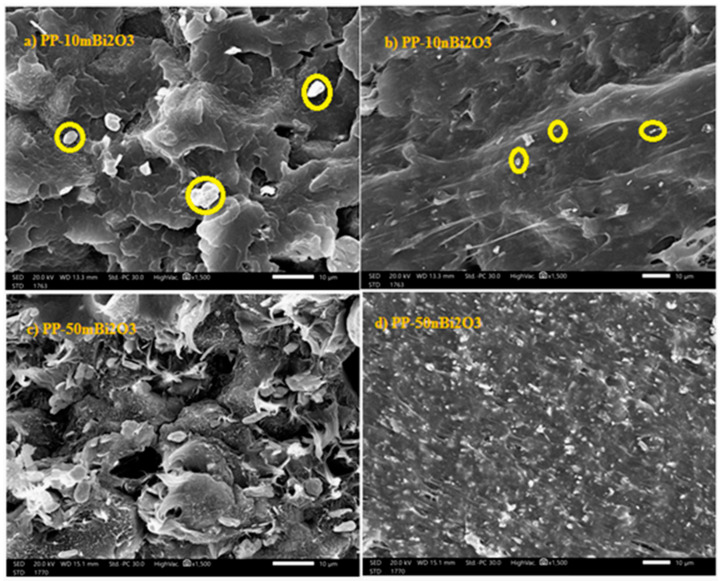
SEM images of micro and nano prepared samples: (**a**) PP-10m Bi_2_O_3_, (**b**) PP-10n Bi_2_O_3_, (**c**) PP-50m Bi_2_O_3_, and (**d**) PP-50n Bi_2_O_3_.

**Figure 4 polymers-14-02253-f004:**
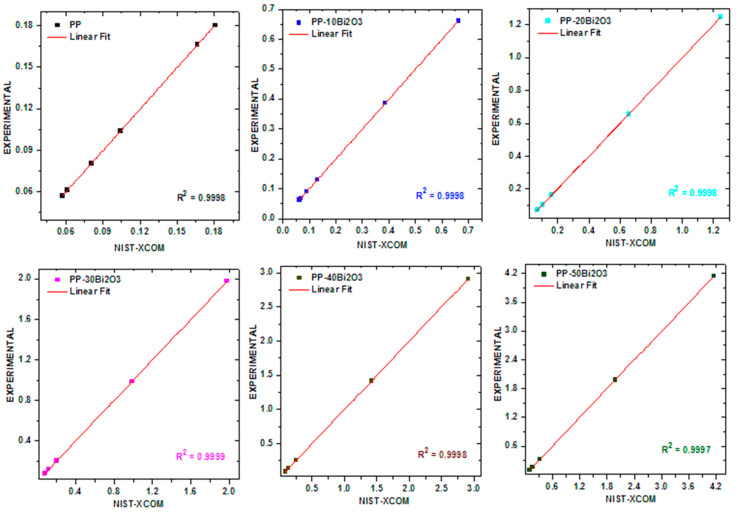
The relation between the experimental and theoretical LAC results.

**Figure 5 polymers-14-02253-f005:**
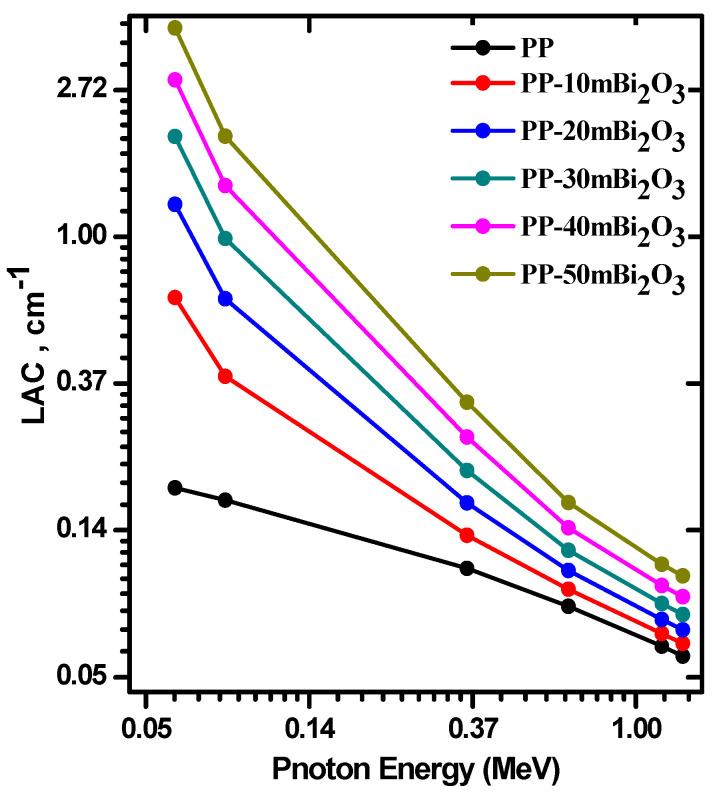
The LAC of pp-micro Bi_2_O_3_ composites as a function of energy.

**Figure 6 polymers-14-02253-f006:**
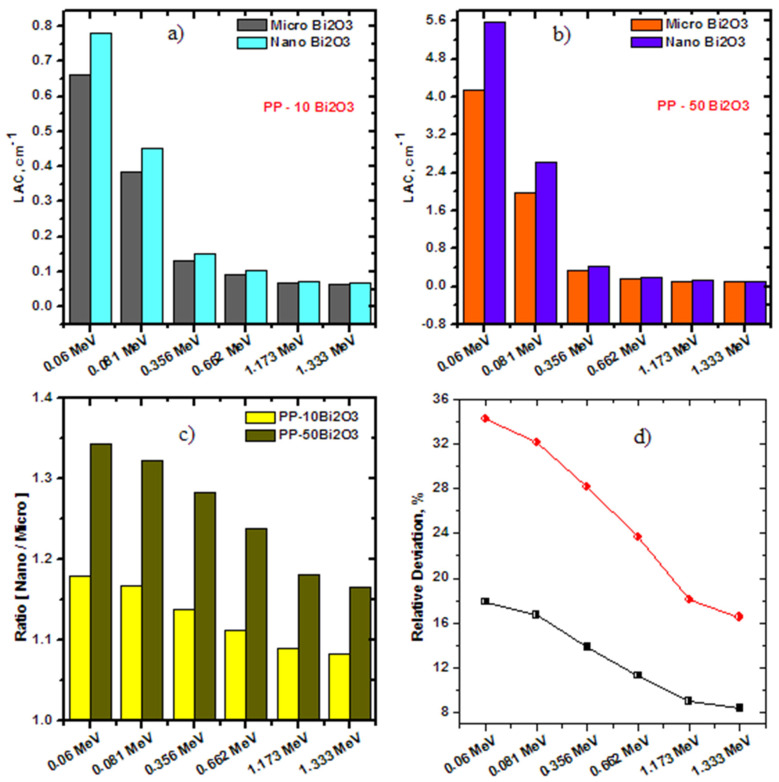
The attenuation comparison between the micro and nano Bi_2_O_3_ as a filler in polypropylene: (**a**) LAC of PP-10m Bi_2_O_3_ and PP-10n Bi_2_O_3_; (**b**) LAC of PP-50m Bi_2_O_3_ and PP-50n Bi_2_O_3_; (**c**) the ratio between the micro and nano filler for both PP-10Bi_2_O_3_ and PP-50Bi_2_O_3_ samples; (**d**) the relative deviation between the micro and nano filler for both PP-10Bi_2_O_3_ and PP-50Bi_2_O_3_ samples.

**Figure 7 polymers-14-02253-f007:**
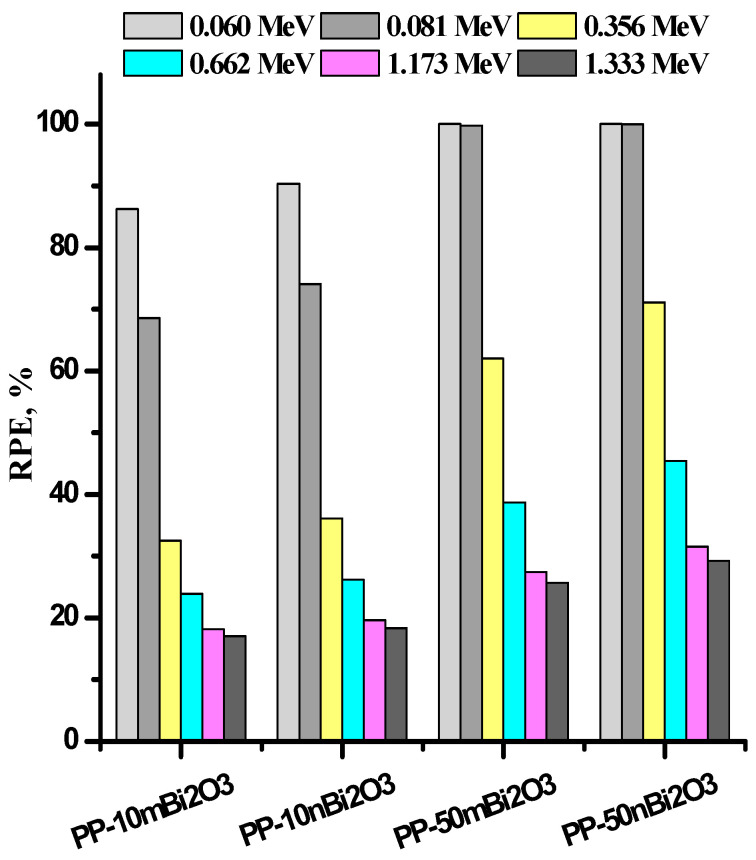
The RPE at different energies for micro and nano polypropylene samples.

**Table 1 polymers-14-02253-t001:** Codes, chemical compositions in weight fraction, and densities of PP-Bi_2_O_3_ composites.

Codes	Compositions (wt%)			Density (g·cm^−3^)
	PP	Bi_2_O_3_		
		Micro	Nano	
**PP**	**100**	—		0.911 ± 0.005
PP-10mPbO10	90	10		1.003 ± 0.004
PP-10nPbO10	90	—	10	1.078 ± 0.009
PP-20mPbO30	80	20	—	1.112 ± 0.009
PP-30mPbO50	70	30	—	1.251 ± 0.006
PP-40mPbO50	60	40	—	1.427 ± 0.003
PP-50mPbO50	50	50	—	1.659 ± 0.008
PP-50nPbO50	50	—	50	1.701 ± 0.006

**Table 2 polymers-14-02253-t002:** The characteristics of the radioactive sources used in this work.

PTB Nuclide	Energy MeV	Emission Probability	Initial Activity Bq	Uncertainty kBq
Am-241	0.060	35.9	259,000	±2.6
Cs-137	0.662	84.99	385,000	±4.0
Ba-133	0.081	32.9	275,300	±1.5
	0.356	62.05		
Co-60	1.173	99.90	212,100	±1.5
	1.333	99.982		

**Table 3 polymers-14-02253-t003:** The half value layer, mean free path, and tenth value layers of prepared micro- and nano-related samples at different energies.

Attenuation Parameters	Energy (MeV)	0.060	0.081	0.356	0.662	1.173	1.333
HVL, cm	PP-10m Bi_2_O_3_	1.048	1.795	5.287	7.609	10.364	11.126
	PP-10n Bi_2_O_3_	0.889	1.538	4.643	6.837	9.507	10.269
	PP-50m Bi_2_O_3_	0.167	0.350	2.147	4.250	6.484	7.001
	PP-50n Bi_2_O_3_	0.124	0.265	1.675	3.435	5.492	6.009
MFP, cm	PP-10m Bi_2_O_3_	1.512	2.590	7.628	10.977	14.952	16.051
	PP-10n Bi_2_O_3_	1.283	2.219	6.699	9.864	13.716	14.816
	PP-50m Bi_2_O_3_	0.241	0.505	3.097	6.131	9.355	10.101
	PP-50n Bi_2_O_3_	0.179	0.382	2.416	4.956	7.923	8.669
TVL, cm	PP-10m Bi_2_O_3_	3.482	5.964	17.564	25.275	34.429	36.960
	PP-10n Bi_2_O_3_	2.954	5.109	15.424	22.713	31.583	34.114
	PP-50m Bi_2_O_3_	0.555	1.162	7.131	14.118	21.540	23.258
	PP-50n Bi_2_O_3_	0.413	0.879	5.564	11.411	18.243	19.961

## Data Availability

All data are available in the manuscript.

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
