# Peer review of "Experimental Study of Polypropylene with Additives of Bi2O3 Nanoparticles as Radiation-Shielding Materials"

_polymers, 2022, doi:10.3390/polym14112253_

Round 1

Reviewer 1 Report

This version does not look worthy and cannot be recommended for publication in this form and at least needs some improvement and clarification.

  1. The choice of bismuth oxide Bi2O3, its actual relevance should be outlined more clearly,
  2. In this regard, possible point defects formed under radiation in Bi2O3 should be discussed and analyzed. As a basis, possible analogies between other similar oxides (Al2O3 or Ga2O3) should be carefully reviewed. See, for example:   

Popov et al. Comparison of the F-type center thermal annealing in heavy-ion and neutron irradiated Al2O3 single crystals. (2018) Nuclear Instruments and Methods in Physics Research, Section B: Beam Interactions with Materials and Atoms, 433, pp. 93-97.

Luchechko A., et al. Shallow and deep trap levels in X-ray irradiated β-Ga2O3: Mg. (2019) Nuclear Instruments and Methods in Physics Research, Section B: Beam Interactions with Materials and Atoms, 441 , pp. 12-17.

  1. Figure 4. Have you checked how it will be modified if the material receives significant doses of radiation. Is it expected to depend on fluence, energy and type of irradiation.
  2. The reference list was made spontaneously and each link is made according to its own format. But after all there are rules of the journal?

Author Response

This version does not look worthy and cannot be recommended for publication in this form and at least needs some improvement and clarification.

Response: We thank the reviewer for his positive comments. Below are the point-by-point responses to comments directed by the reviewer.

  1. The choice of bismuth oxide Bi2O3, its actual relevance should be outlined more clearly,

Reply: Thank you for your remark. The actual relevance of Bi2O3 was clarified in the manuscript.

  1. In this regard, possible point defects formed under radiation in Bi2O3 should be discussed and analyzed. As a basis, possible analogies between other similar oxides (Al2O3 or Ga2O3) should be carefully reviewed. See, for example:   

Popov et al. Comparison of the F-type center thermal annealing in heavy-ion and neutron irradiated Al2O3 single crystals. (2018) Nuclear Instruments and Methods in Physics Research, Section B: Beam Interactions with Materials and Atoms, 433, pp. 93-97.

Luchechko A., et al. Shallow and deep trap levels in X-ray irradiated β-Ga2O3: Mg. (2019) Nuclear Instruments and Methods in Physics Research, Section B: Beam Interactions with Materials and Atoms, 441 , pp. 12-17.

Reply: Thank you for your suggestion. It was discussed in the introduction part.

  1. Figure 4. Have you checked how it will be modified if the material receives significant doses of radiation. Is it expected to depend on fluence, energy and type of irradiation.

Reply: Thank you for your question. Indeed, to test this feature, we exposed these samples to the radiation we have for one week, and nothing changed in the samples. Perhaps with higher energy sources change over the years.

  1. The reference list was made spontaneously and each link is made according to its own format. But after all there are rules of the journal?

Reply: Thank you for your remark. Don't worry about this point. The journal works at this point and arranges the references according to a journal format.

         Finally, we thank the reviewer for his helpful remarks.

Reviewer 2 Report

In this manuscript, PP/Bi2O3 composites were reported with incorporation of micro and nanoparticles of Bi2O3 into PP matrix, and used as radiation shielding materials. The shielding efficiency of gamma rays was tested for different sources with different energies, the results revealed that the PP embedded with nBi2O3 gives better attenuation than its counterpart pp embedded with mBi2O3. However, there are some issues need to be addressed. Therefore, a major revision of this manuscript is recommended.

  • The abbreviation should be illustrated for the first time appeared, such as “LAC, NIST-XCOM”in the abstract. This should be appeared like “half value layer (HVL), Mean Free Path (MFP), Tenth Value Layer (TVL) and Radiation Protection Efficiency (RPE) ”
  • The format of “Bi2O3”in the keywords should be revised.
  • The results of TEM images of Bi2O3 nanoparticles should move to the section of “Results and Discussion”.
  • The format of the paragraph “The prepared samples of PP-mBi2O3 and PP-nBi2O3 were examined using scanning electron microscopy (SEM)...”should be revised according to the context.
  • “PP-10Bi2O3”in the caption of Figure 1 should be changed to “PP-10mBi2O3”.
  • In Figure 1, the resolution of PP-10mBi2O3, PP-50mBi2O3 was not the same as PP-10nBi2O3, PP-50nBi2O3, thus these SEM images is not suitable for the comparison of the dispersion state of mBi2O3 and nBi2O3. SEM with the same resolution for composites embedded mBi2O3 and nBi2O3 should be provided.
  • The format of the references should be revise, and DOI number should be provided according to the journal’s requirement.
  • The shielding efficiency of the obtained composites with the increase of Bi2O3 content, and the maximum content of Bi2O3 reached 50%,which will influence the mechanical properties and thermal stability dramatically. The mechanical properties and thermal stability is very important for the practical application of this materials, therefore, the stress-strain and TG curves should be provided.

Author Response

In this manuscript, PP/Bi2O3 composites were reported with the incorporation of micro and nanoparticles of Bi2O3 into the PP matrix, and used as radiation shielding materials. The shielding efficiency of gamma rays was tested for different sources with different energies, the results revealed that the PP embedded with nBi2O3 gives better attenuation than its counterpart pp embedded with mBi2O3. However, there are some issues need to be addressed. Therefore, a major revision of this manuscript is recommended.

Response: We thank the reviewer for his positive comments. Below are the point-by-point responses to comments directed by the reviewer.

  • The abbreviation should be illustrated for the first time appeared, such as “LAC, NIST-XCOM”in the abstract. This should be appeared like “half value layer (HVL), Mean Free Path (MFP), Tenth Value Layer (TVL) and Radiation Protection Efficiency (RPE) ”

Reply: Thank you for your remark. The abbreviation was illustrated.

  • The format of “Bi2O3”in the keywords should be revised.

Reply: Thank you for your remark. It was revised and corrected

  • The results of TEM images of Bi2O3 nanoparticles should move to the section of “Results and Discussion”.

Reply: Thank you for your remark. It was moved based on your suggestion

  • The format of the paragraph “The prepared samples of PP-mBi2O3 and PP-nBi2O3 were examined using scanning electron microscopy (SEM)...”should be revised according to the context.

Reply: Thank you for your remark. It was revised and the format was corrected.

  • “PP-10Bi2O3”in the caption of Figure 1 should be changed to “PP-10mBi2O3”.

Reply: Thank you for your remark. It was changed

  • In Figure 1, the resolution of PP-10mBi2O3, PP-50mBi2O3 was not the same as PP-10nBi2O3, PP-50nBi2O3, thus these SEM images is not suitable for the comparison of the dispersion state of mBi2O3 and nBi2O3. SEM with the same resolution for composites embedded mBi2O3 and nBi2O3 should be provided.

Reply: Thank you for your remark. SEM with the same resolution for composites embedded mBi2O3 and nBi2O3 was provided

  • The format of the references should be revise, and DOI number should be provided according to the journal’s requirement.

Reply: Thank you for your remark. Don't worry about this point. The journal works at this point and arranges the references according to a journal format.

  • The shielding efficiency of the obtained composites with the increase of Bi2O3 content, and the maximum content of Bi2O3 reached 50%,which will influence the mechanical properties and thermal stability dramatically. The mechanical properties and thermal stability is very important for the practical application of this materials, therefore, the stress-strain and TG curves should be provided.

Reply: Thank you for your suggestion. We are already working on mechanical and thermal measurements, and the reason why they are not put into this work is because in the future manuscript, the mechanical and thermal properties before and after exposure of these materials to gamma rays will compared and discussed, and whether or not photons will affect these properties will be discussed. Therefore, we decided to study the shielding properties of these samples first to be taken as a reference for the next research. I hope you understand that, and on the other hand, if there are any other additions from your side, we will be ready to deal with them.

Finally, we thank the reviewer for his helpful remarks.

Round 2

Reviewer 1 Report

The authors have significantly revised the original manuscript so that it can now be recommended for publication

Author Response

Thank you for your reviewing

Reviewer 2 Report

Though the author made a revision of this manuscript, yet most of the issues were not addressed.Therefore, a a major revision of this manuscript is recommended.

(1) The results of TEM images of Bi2O3 nanoparticles were not move to the section of “Results and Discussion”.

(2) SEM with the same resolution for composites embedded mBi2O3 and nBi2O3 were not provided. As shown in the SEM image, the resolution for the SEM of composites embedded mBi2O3 is x1500,however, the resolution for the SEM of composites embedded nBi2O3 is x1000. The resolution of PP-10mBi2O3, PP-50mBi2O3 was not the same as PP-10nBi2O3, PP-50nBi2O3, thus these SEM images is not suitable for the comparison of the dispersion state of mBi2O3 and nBi2O3. 

(3) DOI number of the references were not provided according to the journal’s requirement.

(4) The maximum content of Bi2O3 reached 50%, which will influence the mechanical properties and thermal stability dramatically. The mechanical properties and thermal stability is very important for the practical application of this materials, however, the stress-strain and TG curves were not provided.

Author Response

The results of TEM images of Bi2O3 nanoparticles should move to the section of “Results and Discussion”.

Reply: It was moved

SEM with the same resolution for composites embedded mBi2O3 and nBi2O3 were not provided. As shown in the SEM image, the resolution for the SEM of composites embedded mBi2O3 is x1500,however, the resolution for the SEM of composites embedded nBi2O3 is x1000. The resolution of PP-10mBi2O3, PP-50mBi2O3 was not the same as PP-10nBi2O3, PP-50nBi2O3, thus these SEM images is not suitable for the comparison of the dispersion state of mBi2O3 and nBi2O3. 

Reply: All  figures have the same resolution.

DOI number of the references were not provided according to the journal’s requirement. Reply:  It was added

The maximum content of Bi2O3 reached 50%, which will influence the mechanical properties and thermal stability dramatically. The mechanical properties and thermal stability is very important for the practical application of this materials, however, the stress-strain and TG curves were not provided.

Reply: It will be discussed in the next work. The shielding , mechanical and thermal properties of both the micro and nano (Not Micro only) samples cannot be presented in the same article, and should be divided into at least two materials.

Round 3

Reviewer 2 Report

Corrections were performed, the manuscript has been improved and now it is suitable for publishing